# Coping with Water Stress: Ameliorative Effects of Combined Treatments of Salicylic Acid and Glycine Betaine on the Biometric Traits and Water-Use Efficiency of Onion (*Allium cepa*) Cultivated under Deficit Drip Irrigation

**DOI:** 10.3390/biom13111634

**Published:** 2023-11-09

**Authors:** Muziri Mugwanya, Fahad Kimera, Anwar Abdelnaser, Hani Sewilam

**Affiliations:** 1Center for Applied Research on the Environment and Sustainability (CARES), School of Science and Engineering, The American University in Cairo, AUC Avenue, P.O. Box 74, New Cairo 11835, Egypt; muziri@aucegypt.edu (M.M.); fkim@aucegypt.edu (F.K.); 2Institute of Global Public Health, School of Science and Engineering, The American University in Cairo, AUC Avenue, P.O. Box 74, New Cairo 11835, Egypt; anwar.abdelnaser@aucegypt.edu; 3UNESCO Chair in Hydrological Changes and Water Resources Management, RWTH Aachen University, 52062 Aachen, Germany

**Keywords:** water stress, deficit drip irrigation, osmoprotectants, plant growth regulators

## Abstract

Freshwater scarcity is a major global challenge threatening food security. Agriculture requires huge quantities of water to feed the ever-increasing human population. Sustainable irrigation techniques such as deficit drip irrigation (DDI) are warranted to increase efficiency and maximize yield. However, DDI has been reported to cause water stress in plants. The study aimed to investigate the influence of the exogenous application of salicylic acid alone (SA) or in combination with glycine betaine (GB) on the growth, yield quality, and water-use efficiency of onions under different DDI treatments (100%, 70%, and 40% field capacity (FC)). Spray treatments (sub-treatments) were as follows: T1: (distilled water), T2: (1.09 mM SA), T3: (1.09 mM SA + 25 mM GB), T4: (1.09 mM SA + 50 mM GB), and T5: (1.09 mM SA + 100 mM GB). Our results indicated that T2 slightly ameliorated the effects of water stress by improved plant heights, leaf number, pseudostem diameter, bulb quality, and nutrient content of onion bulbs, especially under the 70% FC treatment. However, T3 recorded the poorest results on leaf number, pseudostem diameter, and bulb quality under the 70% and 40% FC treatments. Generally, our results indicated that onions could tolerate moderate water stress (70% FC) without severely affecting the growth and yield of onion. In conditions where freshwater is a limiting factor, a DDI treatment of 40% FC is recommended.

## 1. Introduction

The ever-increasing global population and fast-growing world economies have resulted in increased food demand amidst freshwater scarcity [1,2]. According to the European Environmental Agency (EEA), water scarcity is one of the major world concerns that should be tackled for the continuity of life on our planet [3]. An estimated 5.7 billion people will be directly impacted by freshwater scarcity by 2050 [2,4]. The Middle East and North Africa regions constitute the most highly at-risk countries from the current global water crises. This indicates that a quarter of the world’s population will be vulnerable to water shortages in the coming few years [5]. For instance, Egypt, with the third-largest population in Africa, currently faces a freshwater shortage of 13.5 billion cubic meters per year, and by 2025, this is estimated to reach 26 billion cubic meters [6]. Freshwater scarcity has a direct impact on food production [7]. Fresh fruit and vegetable production requires large amounts of water input in the production chain process [8]. Any water shortages in their production chain will negatively impact the industry and thus lead to substantial economic losses.

Dry onions (*Allium cepa*) are one of the most commonly cultivated vegetable crops worldwide. According to Statista [9], the global production of dry onions by 2020 stood at 104.55 million metric tons, making dry onions the second most widely cultivated vegetable crop after tomatoes. Egypt is among the top ten 2020 producers of dry onions worldwide with a total harvested area of ~89,018 tons and a total production of ~3.1 million metric tons [10]. However, the current freshwater crisis in Egypt and the use of unsustainable irrigation techniques such as flood irrigation threaten its production. Water conservation irrigation techniques such as deficit drip irrigation (DDI) are effective practices in reducing the amount of irrigation water supplied while increasing the water-use efficiency (WUE) of plants [11]. It is imperative to note that DDI requires precise knowledge of the crop’s water requirements at the different stages of its growth cycle and its yield response regarding the amount of water applied to optimize the crop’s productivity and its physiological processes [11,12,13]. Several studies have shown improved WUE, yield, and fruit quality of several plant species cultivated under DDI conditions [14,15,16,17,18,19]. However, DDI can trigger water stress which can lower plant growth and yield. For example, Basal et al. [20] investigated the effects of different DDI ratios (Irrigation rate (IR)-0, IR-25, IR-50, IR-75, and IR-100) on the yield and WUE of cotton plants and observed reduced yields and WUE in IR-25 and IR-50 compared to IR-70 and IR-100. Abdelkhalik et al. [21] demonstrated that reducing the water applied to 75% of the irrigation water requirement (IWR) of pepper (*Capsicum annuum* L.) at the harvesting stage resulted in a reduction in crop yield. Moreover, higher incidences of blossom end rot were noted in plants irrigated with 75% or 50% IWR during the entire growth cycle, thus leading to a decline in the marketable fruit yield. In another study, Parkash et al. [22] reported reduced plant growth and fruit yields in cucumbers cultivated under subsurface DDI conditions of 60% and 40% crop evapotranspiration (ETc) compared to those irrigated at 80% and 100% ETc.

Several interventions have been implemented to enhance crop production under DDI conditions to manage water stress effects in plants. Exogenous application of plant growth regulators (PGRs) and osmoprotectants has been reported to be an effective approach to ameliorating the negative effects of several abiotic stressors in different plant species cultivated under different DDI treatments [23,24,25,26,27,28]. PGRs are naturally synthesized phytohormones that play diverse roles in plant growth and protection [29]. Salicylic acid (SA) is among the most widely studied and exogenously applied PGR. It plays a vital role in several plants’ physiological and biochemical processes such as growth, development, membrane permeability, photosynthesis, flower induction, ion uptake, and enzymatic activities [29,30,31]. Under plant abiotic stress, SA acts as a signaling molecule to induce the expression of stress-responsive genes and biosynthesis of stress-related proteins thus triggering several morphological and physiological responses [32]. On the other hand, osmoprotectants are low-molecular-weight organic compounds that are highly soluble and electrically neutral with no cell toxicity at high concentrations [33,34]. They protect plant cells from stress via the protein and membrane stabilization of cells to prevent the dehydration of tissues. Furthermore, their accumulation in plant tissues facilitates the maintenance of both the osmotic and turgor pressure of cells as well as scavenging for reactive oxygen species (ROS) [29,35,36]. Glycine betaine (GB) is among the most widely studied and exogenously applied osmoprotectants with a great potential for ameliorating several abiotic stress factors in plants [37,38,39,40,41,42,43]. Exogenous application of GB in plants leads to the expression of stress-related genes that are involved in both biotic and abiotic stress tolerance [44]. To the best of our knowledge, no studies have so far been conducted on the combined effects of the exogenous application of SA and GB on onion cultivated under different DDI treatments. The aim of this study, therefore, was to investigate the influence of SA alone or in combination with different concentrations of GB on the growth, yield production, and the biochemical and nutritive composition of onion under different DDI treatments.

## 2. Materials and Methods

### 2.1. Experimental Layout

A field experiment was conducted during the winter season of 2020/2021 at the Center for Applied Research on the Environment and Sustainability (CARES), the American University in Cairo (AUC), New Cairo Egypt (30°01′11.7″ N 31°29′59.8″ E). The experimental layout was a split-plot design with three replications. The main plots were the different deficit drip irrigation (DDI) treatments and the subplots were the exogenous spray treatment applications. Healthy and homogeneously grown seedlings were transplanted on 1 November 2020 at an inter- and intra-row spacing of 15 cm and 70 cm, respectively. Seedlings of onions (Giza 40 variety) were obtained from the Agricultural Research Center in Cairo, Egypt.

Fertilization and pest control were conducted according to the recommendations of the Egyptian ministry of agriculture. The average minimum and maximum temperatures during the growing season were 13.7 °C and 26.5 °C, respectively. Likewise, the average precipitation, grass reference evapotranspiration (ETo), wind speed, and downward radiation during the growing season were 2.9 mm, 136.1 mm, 2.8 mps, and 223.4 (W/m^2^), respectively.

### 2.2. Soil Sample Analysis

Soil samples were randomly obtained from different spots of the field at the depth of 5 cm and pooled for chemical and physical analysis and brought to the Agricultural Research Center (ARC), Giza Egypt, for analysis. Results of the soil analysis are presented in Table 1. According to the recommendations from the Egyptian Ministry of Agriculture, compost was mixed in the soil at the rate of 20 m^3^/feddan for onion production in sandy soils.

### 2.3. Experimental Treatments

A soil moisture sensor (TDR 350 from Specmeters, Aurora, IL, USA) was used to monitor the electro-conductivity (EC) and soil moisture content twice a week and DDI treatments (main treatments) were applied at the following field capacity (FC): 100% FC, 70% FC, and 40% FC. DDI treatments were initiated 45 days after transplanting (DAT) when the plants had been fully established in the soil.

Sub-treatment concentrations of 1.09 mM SA, 25 mM GB, 50 mM GB, and 100 mM GB were prepared and the pH was adjusted to 7.0. A surfactant tween 20 (0.5%) was added to the prepared solutions as a wetting agent. The selection of SA concentration was based on previous studies [45,46,47], which indicated its efficiency in increasing abiotic stress tolerance in onions. The exogenous spray treatments were as follows: T1: distilled water (control); T2: 1.09 mM SA + 0 mM GB; T3: 1.09 mM SA + 25 mM GB; T4: 1.09 mM SA + 50 mM GB; T5: 1.09 mM SA + 100 mM GB. Exogenous application of the sub-treatments was performed 3 times (i.e., at 30, 45, and 60 DAT). All the chemicals used in this study were purchased from the Alpha chemika company in Cairo, Egypt.

### 2.4. Growth Parameter Measurements

Six uniformly growing plants from each sub-treatment were randomly tagged for sampling. Growth parameter measurements (plant height, leaf number, and pseudostem diameter) were obtained at 60, 75, and 90 DAT. Plant height was measured from the base of the plant to the terminal growing point of the leaves using a meter scale; leaf number was obtained by counting the number of leaves per plant and averages were determined. Pseudostem diameter was measured using a hand-held digital vernier caliper from Mitutoyo, Takatsu-ku, Japan.

### 2.5. Irrigation Water-Use Efficiency (IWUE)

As reported by Mugwanya et al. [29], the IWUE was calculated by the following equation:IWUE = (Y/I)
where IWUE = irrigation water-use efficiency; Y = yield (kg/ha); I = applied amount of water (m^3^).

### 2.6. Bulb Quality and Yield Measurements

After harvesting, dry onion bulbs per sub-treatment were graded according to size as follows: Grade A (<50 mm), Grade B (50–75 mm), and Grade C (>75 mm). Polar and equatorial bulb diameters were measured using hand-held digital vernier calipers and averages were determined. Furthermore, the percentage of bulb doubles from each sub-treatment was calculated. For fresh bulb yield, 6 onion bulbs per each sub-treatment were randomly obtained and weighed and yield was expressed as ton/ha.

### 2.7. Bulb Biochemical Analysis

Quantification of Vitamin C: Vitamin C was determined according to the method described by Desai and Desai [48]. Briefly, 5 g of the onion sample was blended and the sample was mixed with 25 mL of 5% metaphosphoric acid acetic acid solution. The remaining amount of 25 mL of 5% metaphosphoric acid acetic acid solution was added and the solution was filtered using a Whatman filter paper. A few drops of bromine water were added to the solution, followed by the addition of a few drops of thiourea solution until the solution was clear. Next, 1 mL of 2,4 DNPH solution was added to the solutions and all the prepared standard solutions for the determination of the calibration curve, and then, the samples were incubated at 37 °C for 3 h. The solution was then cooled on ice and 2.5 mL of sulfuric acid was added. Colored solutions were obtained whose absorbance was measured at 521 nm against the blank using a spectrophotometer (Jenway, Sungai Petani, Malaysia, 6305). Vitamin C concentrations were calculated from the standard curve.

Quantification of total carbohydrates: The total carbohydrates were quantified according to the method described by Silkina et al. [49]. Briefly, 1 g of the onion sample was dipped in liquid nitrogen and crashed with a motor and pestle to form a powder. The sample was then stored at −40 °C until further analysis. Before analysis, samples were removed from the freezer and allowed to warm at room temperature, and placed in a desiccator to remove any water and prevent any hydrating. Then, 5 mg of the sample was weighed into boiling test tubes and 2 mL of 1 M sulfuric acid was added to each sample. The samples were then heated for 1 h at 90 °C in a water bath. The samples were then centrifuged for 10 min at 5500 rpm. Next, 0.1 mL of the supernatant was pipetted into the boiling test tubes, followed by 2.5 mL of concentrated sulfuric acid, followed by 0.5 mL of phenol. The samples were then kept at room temperature for 20 min and carefully poured into acid-resistant cuvettes and absorbance was recorded at 485 nm against the blank using a spectrophotometer (Jenway 6305). Total carbohydrate concentrations were calculated from the standard curve.

### 2.8. Microwave Digestion and Bulb Nutritive Composition Analysis

Nutritive composition analysis was performed as described by Ariyama et al. [50]. Triplicate samples of 300 mg of fresh onion bulbs were obtained and dried in an oven at 70 °C for 2 h, followed by nitric acid (69%) digestion with a microwave (ENTRY SW-E). Certified reference material matching the sample matrix as well as blank samples were included in each run. The digested samples were subsequently diluted to 50 mL with ultrapure water (Milli-Q Element, Millipore, Burlington, MA, USA) and stored at room temperature until analysis. Nutrient composition analysis was performed with inductively coupled plasma–optical emission spectrometry (ICP-OES). All samples were analyzed undiluted using an ICP-OES (Optima 5300 DV, PerkinElmer, Hong Kong, China) equipped with a Meinhard nebulizer and a cyclonic spray chamber. Instrument settings were as follows: RF power, 1400 W; nebulizer flow, 0.65 L/min; auxiliary flow, 0.2 L/min; plasma flow, 15 L/min; sample flow, 1.5 mL/min. Interference-free wavelengths were chosen for all elements and used in either axial or radial mode. External calibration was conducted using two commercially available standard solutions (P/N 4400-132565 and P/N 4400-ICP-MSCS, CPI International, Amsterdam, The Netherlands). ICP-OES data were processed with Winlab32 software (version 3.1.0.0107, PerkinElmer). The analytical accuracy was evaluated using four different reference materials, namely, white cabbage (BCR-679, Institute for Reference Materials and Measurements, Geel, Belgium), apple leaves, and durum wheat flour (NIST 1515 and NIST 8436, respectively, National Institute of Standards and Technology, Gaithersburg, MD, USA), and leek (in-house standard, FoodDTU, Lyngby, Denmark). Full-quantitative data were rejected if below the limit of detection (LOD). LOD was determined as 3 times the standard deviation of at least seven blanks. Furthermore, data were rejected if the accuracy for each element was <90% of the reference value. As such, the following macro- and microelements were analyzed (K, Ca, Mg, Mn, Cu, Zn, Fe).

### 2.9. Statistical Analysis

The collected data were analyzed using IBM-SPSS Statistical Tool (Version 22) and expressed as mean ± standard deviation (SD). Normality and Levene’s tests were conducted to determine whether the data followed a normal distribution and fulfilled the equality of variances for the calculation of Analysis of Variance (ANOVA). ANOVA (both one-way and two-way) was performed to detect significant differences in all the measured parameters and the difference in means analyzed by Tukey’s Honest Significant Difference (HSD) test at α = 0.05. A principal component analysis (PCA) was performed to explore possible similarities and differences in the nutrient composition of onion bulbs cultivated under different deficit drip irrigation treatments and different concentrations of exogenously applied osmoprotectants and plant growth regulators. The PCA was performed using the ‘prcomp’ function and the generated plots were visualized using the ‘ggbiplot’ package of R Statistical Programming Language (version 4.1.0).

## 3. Results

### 3.1. Plant Growth Parameters

The results of different plant growth parameters (plant height, leaf number, and pseudostem diameter) are presented in Table 2. No significant differences in plant height were noted among all the sub-treatments (T1, T2, T3, T4, and T5) at 100% and 40% FC across all sampling time-points (60, 75, and 90 DAT). However, at 70% FC, T2 significantly recorded higher values for plant height at 75 and 90 DAT compared to other sub-treatments (*p* < 0.05). Generally, plants irrigated at 70% FC recorded significantly higher values for plant height compared to those irrigated at 40 and 100% FC (*p* < 0.05).

Data on leaf number per plant indicated no significant differences among all the sub-treatments in plants irrigated at 100%, 70%, and 40% FC (75 and 90 DAT). Generally, plants irrigated at 100% and 70% FC recorded significantly higher values for leaf number per plant compared to those irrigated at 40% FC (*p* < 0.05).

For pseudostem diameter, no significant differences were noted among all the sub-treatments at 100%, 70%, and 40% FC across all the sampling time-points (60, 75, and 90 DAT). Still, the plants irrigated at 100% and 70% FC recorded significantly higher pseudostem diameter values than those irrigated at 40% FC (*p* < 0.05).

### 3.2. Fresh Bulb Yield and Irrigation Water-Use Efficiency (IWUE)

Results of fresh bulb yield and irrigation water-use efficiency are presented in Figure 1. No significant differences in fresh bulb yield were noted among all the sub-treatments at 100%, 70%, and 40% FC. However, higher fresh bulb yields were noted in plants irrigated at 70% and 40% FC compared to those irrigated at 100% FC (*p* < 0.05) (Figure 1a). The results of the IWUE are presented in Figure 1b. No significant differences in IWUE were noted among all sub-treatments in plants irrigated at 100%, 70%, and 40% FC. Overall, 40% FC recorded, significantly, the highest values for IWUE compared to 70% and 100% FC (*p* < 0.05).

### 3.3. Bulb Equatorial and Polar Diameter

The results on bulb equatorial and polar diameter are presented in Figure 2a,b, respectively. No significant differences in the bulb equatorial diameter were noted among all the sub-treatments. However, plants irrigated at 70% FC recorded significantly higher values for the bulb equatorial diameter compared to those at 100% FC (*p* < 0.05). Similarly, no significant differences in bulb polar diameter were noted among all sub-treatments in plants irrigated at 100%, 70%, and 40% FC. However, 70% FC recorded significantly higher values for bulb polar diameter compared to 40% and 100% FC (*p* < 0.05).

### 3.4. Bulb Biochemical Composition

Results on the biochemical composition of onion bulbs are presented in Figure 3. As shown in Figure 3a, T1 recorded, significantly, the highest values for vitamin C content compared to other sub-treatments in plants irrigated at 100% and 70% FC (*p* < 0.05). At 40% FC, T1 recorded, significantly, the lowest values for vitamin C content compared to other sub-treatments (*p* < 0.05). Overall, 100% FC recorded, significantly, the highest vitamin C content compared to 70% and 40% FC. For total carbohydrate concentration, no significant differences were noted among all sub-treatments in plants irrigated at 100%, 70%, and 40% FC (Figure 3b). Overall, 40% and 70% FC recorded the higher values for carbohydrate content compared to 100% FC.

### 3.5. Bulb Quality

Bulb quality, in this case, is measured as the desirable market size as well as the incidences of having either a double bulb or single bulb; this is presented in Table 3. The results indicated no significant differences in percentage doubles among all sub-treatments in plants irrigated at 100%, 70%, and 40% FC. However, a significantly higher incidence of percentage doubles was recorded at 70% FC compared to 100% FC (*p* < 0.05) (Table 3). Furthermore, the results presented in different grading groups (Grade A, B, and C) of onion bulbs indicated no significant differences among all the sub-treatments (Table 3). However, 40% FC recorded a significantly lower percentage of Grade A onion bulbs compared to 100% FC (*p* < 0.05), whereas 70% FC recorded a significantly lower percentage of Grade B onion bulbs compared to 40% and 100% FC (*p* < 0.05). For onion bulbs in Grade C, 70% FC recorded significantly higher percentage values, followed by 40% and 100% FC (*p* < 0.05).

### 3.6. Bulb Nutrient Composition and Principal Component Analysis

The results of the macroelement composition in fresh onion bulbs are presented in Table 4. Data on the potassium (K) content of the bulbs indicated that T5, T4, and T3 recorded, significantly, the highest K content in onion bulbs at 100%, 70%, and 40%, respectively (*p* < 0.05). Overall, 40% FC recorded, significantly, the highest K content in onion bulbs compared to 70% and 100% FC (*p* < 0.05). For calcium (Ca) content, T4 and T3 recorded, significantly, the highest Ca content compared to other sub-treatments at 100%, 70%, and 40% FC (*p* < 0.05). Overall, 70% FC recorded significantly higher values for Ca content in onion bulbs compared to 40% and 100% FC (*p* < 0.05). For magnesium (Mg) content, T2 and T5 recorded significantly higher Mg content in onion bulbs compared to other sub-treatments at 40%, 70%, and 100% FC (*p* < 0.05). Overall, 40% FC recorded higher values for the Mg content of onion bulbs compared to 70% and 100% FC (*p* < 0.05).

The results of the microelement composition in fresh onion bulbs are presented in Table 5. Data on the manganese (Mn) content in onion bulbs showed that T1 and T4 recorded significantly lower Mn contents in onion bulbs compared to other sub-treatments at 100%, 70%, and 40% FC (*p* < 0.05). Overall, 40% FC recorded higher values for the Mn content of onion bulbs compared to 70% and 100% FC (*p* < 0.05). For copper (Cu) content, T1, T3, and T5 recorded, significantly, the highest values compared to other sub-treatments at 70% and 100% FC (*p* < 0.05). No significant differences in Cu content were noted across all sub-treatments at 40% FC.

Overall, the plants irrigated at 100% and 40% FC recorded significantly higher values for the Cu content of onion bulbs compared to those irrigated at 70% FC (*p* < 0.05). The results on the zinc (Zn) content of onion bulbs indicated that T2 and T3 recorded significantly higher values compared to other sub-treatments at 100%, 40%, and 70% FC. Overall, plants irrigated at 100% FC recorded significantly higher values for Zn content in onion bulbs, followed by 40% and 70% FC (*p* < 0.05). For iron (Fe) content, T5 and T2 recorded significantly higher values compared to other sub-treatments at 100%, 40%, and 70% FC. Overall, plants irrigated at 40% FC recorded significantly higher values for Fe content in onion bulbs compared to those irrigated at 70% and 100% FC (*p* < 0.05).

Principle component analysis (PCA) of nutrient composition data indicated that different sub-treatments influenced the concentration of nutrients in onion bulbs (Figure 4a). And as such, T2 induced the highest concentration of nutrients, followed by T1, T3, T4, and T5. Data on the main treatments (DDI) showed that the 40% FC had a higher concentration of nutrients, followed by 100% and 70% FC (Figure 4b). Data on the distribution of macro- and micronutrients under different DDI treatments showed an increase in potassium (K) concentration in 40% and 70% FC, whereas the concentrations of calcium (Ca), copper (Cu), and iron (Fe) decreased (Figure 4c). Furthermore, the PCA indicated medium and strong correlations among nutrients (Figure 4d).

## 4. Discussion

The present study indicated that irrigation water shortage in the form of DDI has a negative impact on the vegetative growth parameters (plant height, leaf number, and pseudostem diameter) of onions. Overall, 40% FC recorded, significantly, the lowest values for plant height, leaf number, and pseudostem stem diameter relative to 70% and 100% FC. Plant water stress causes chlorophyll degradation, reducing photosynthesis and materialization, leading to growth retardation [51]. A previous study by Shirzadi et al. [52] showed that cultivating onion plants under water stress (60% water requirement) led to a decline in plant height and leaf number per plant. Almaroai and Eissa [53] also reported that exposing onion plants to water stress (50% available water) led to a significant decline in plant height, leaf number, and leaf area. Likewise, Semida et al. [27] observed a significant reduction in vegetative growth (plant height, leaf number per plant, leaf area per plant, and shoot dry weight per plant) of onion cultivated under conditions of water stress (60% ETc). In this study, however, 70% FC recorded higher values for vegetative growth parameters relative to 40% FC, indicating that onion can be grown under moderate water stress conditions without severely impacting vegetative growth. Exogenous application of SA alone or in combination with GB did not induce any effect on the leaf number per plant and pseudostem diameter except for plant heights. Plants sprayed with SA alone (T2) significantly recorded higher values for plant height compared to T1 (control). Similar results have been reported in several crops such as corn [54,55,56], tomato [57,58], cucumber [59,60], and pepper [61,62,63] cultivated under abiotic stress conditions. SA induces antioxidative capacity in plants (i.e., increase in catalase and superoxide dismutase activities) under water stress conditions, partially protecting membrane integrity [64,65,66,67].

Different DDI treatments influenced the fresh bulb yield, whereas no significant differences were noted among the different sub-treatments. Plants irrigated at 70% FC (moderate water stress) generally recorded higher yields, followed by those irrigated at 40% and 100% FC. Our results are in agreement with those reported by Abdelkhalik et al. [68]. The authors demonstrated that cultivating onion plants under water stress (i.e., 75% irrigation water requirement at the bulb ripening stage) resulted in satisfactory bulb yield and increased water-use efficiency. However, the results of our study indicated lower values for IWUE in plants irrigated at 70% and 100% FC compared to those irrigated at 40% FC. The discrepancy in results could be attributed to the difference in experimental conditions such as plant variety, the stage at which water stress was applied, and growth conditions. It is imperative to note that it is more profitable for farmers to maximize IWUE than yield in conditions of freshwater scarcity [52,69]. An interesting finding of this study is that the DDI treatment of 40% FC significantly increased the IWUE by 73.30% compared to 70% and 100% FC. However, when combined with the exogenous application of T2 (DDI, 40% FC) and T3 (DDI, 70% FC), there was a slight improvement in IWUE by 0.57% and 3.73%, respectively, although not significantly different from other sub-treatments. Enhancing the IWUE in response to the application of PGRs and osmoprotectants modulates the stomatal opening and improves the leaf water content of plants which together result in better use of water by the plant. Zali and Ehsanzadeh [70] demonstrated that the exogenous application of 20 mM proline in fennel plants cultivated under drought stress (35–45% or 75–85% of the soil available water depletion) increased their water-use efficiency (WUE). Ahmed et al. [71] investigated the influence of the exogenous application of GB (50 mL/m^2^) on the WUE of winter wheat (*Triticum aestivum* L.) cultivated under conventional and limited irrigation conditions and observed improved WUE in plants treated with GB under both growth conditions. Similarly, Abbaszadeh et al. [72] observed that cultivating *Rosmarinus officinalis* L. under drought stress (60% FC) combined with the exogenous application of 1 mM SA led to improved yields and WUE in plants.

Water stress at specific stages of onion development can lead to a reduction in bulb quality, leading to the development of multi-centered bulbs and reduced bulb diameter [73]. In this study, plants irrigated at 70% FC recorded, significantly, the highest values for percentage doubles (multi-centered bulbs) compared to those at 40% and 100% FC. Our results are in agreement with those reported by Shock et al. [74]. Late water stress application and warm weather seem to increase the incidence of multi-centered bulbs, thus affecting the marketable yield of onion.

Several biochemical changes occur during water or drought stress in plants. Ascorbic acid (vitamin C) is one of the most abundant water-soluble vitamins present in plant tissues. Exposing plants to water stress can affect the biosynthesis of vitamin C [75]. In this study, there was a 40% and 52.47% decline in the vitamin C content in onion bulbs irrigated at 70% and 40% FC, respectively. Such water stress-induced reductions in vitamin C content in plant tissues have also been reported in soybean [75], wheat [76], and several *Labiatae* species [77]. However, the results of this study showed an increased accumulation of total carbohydrates in plants irrigated at 70% and 40% FC. Similar results have been previously reported in apple [78], wheat [79], tomato [80], and Edamame (*Glycine max*. L. Merrill) [81] cultivated under drought stress conditions. Accumulation of total carbohydrates under drought or water stress is a defensive mechanism in plants to prevent oxidative stress cellular damage, thus enhancing plant tolerance toward abiotic stress [82,83]. The nutritive composition of onion bulbs varied across different sub-treatments with T2 (SA alone), recording significantly higher values for the nutritive composition in onion bulbs compared to other sub-treatments (Figure 4a). SA has been previously reported to enhance nutrient uptake in plants [84,85]. Likewise, the results of the PCA indicated a higher accumulation of both macro- and micronutrients in plants irrigated at 40% FC compared to those at 70% and 100% FC. This could be attributed to several factors such as the reduced absorption of soil moisture and increasing plant tolerance to abiotic stress. For instance, increased concentrations of K^+^ were observed in 40% and 70% FC treatment groups. Potassium has been previously reported to enhance drought tolerance in plants by inducing changes in plant root structure, influencing the release of root exudates, and changes in microbial diversity within the soil profile [86]. Furthermore, K^+^ plays a vital role in the regulation of the opening and closing of stomatal guard cells, hence minimizing water loss during water stress [87].

## 5. Conclusions

In conclusion, this study demonstrated that onions could tolerate moderate water stress (70% FC) without severely affecting the vegetative growth and yield of onions. The exogenous application of T2 (SA alone) slightly ameliorated the water stress effects in plants as indicated by improved vegetative growth (plant heights), reduced incidents of percentage doubles, and increased accumulation of both macro- and micronutrients in onion bulbs. In conditions where freshwater is a limiting factor, a DDI treatment of 40% FC is recommended to achieve maximum irrigation water-use efficiency.

## Figures and Tables

**Figure 1 biomolecules-13-01634-f001:**
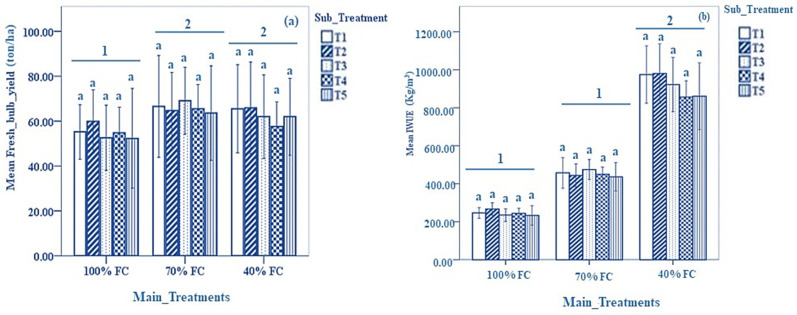
Results of (**a**): fresh bulb yield and (**b**): irrigation water-use efficiency under different DDI treatments and sub-treatments. Data expressed as mean ± SD. Error bars represent the standard deviation. Bar columns having the same letter under the same DDI treatment indicate no significant differences at *p* < 0.05. Horizontal lines at the top of bar columns having different numbers at the top show a significant difference between the DDI treatments at *p* < 0.05. Sub-treatments T1: control (distilled water); T2: 1.09 mM salicylic acid + 0 mM glycine betaine; T3: 1.09 mM salicylic acid + 25 mM glycine betaine; T4: 1.09 mM salicylic acid + 50 mM glycine betaine; T5: 1.09 mM salicylic acid + 100 mM glycine betaine; FC: filed capacity.

**Figure 2 biomolecules-13-01634-f002:**
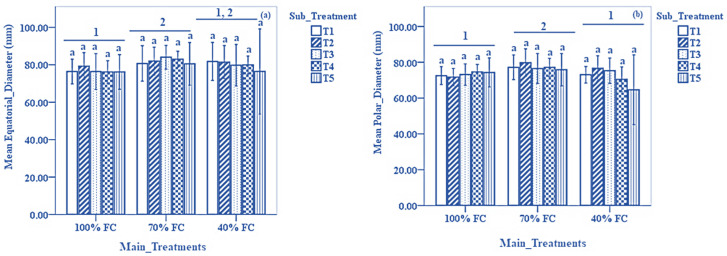
Results of (**a**): average equatorial diameter and (**b**): polar diameter of onion bulbs under different DDI treatments and sub-treatments. Data expressed as mean ± SD. Error bars represent the standard deviation. Bar columns having the same letter under the same DDI treatment indicate no significant differences at *p* < 0.05. Horizontal lines at the top of bar columns having different numbers at the top show a significant difference between the DDI treatments at *p* < 0.05. Sub-treatments T1: control (distilled water); T2: 1.09 mM salicylic acid + 0 mM glycine betaine; T3: 1.09 mM salicylic acid + 25 mM glycine betaine; T4: 1.09 mM salicylic acid + 50 mM glycine betaine; T5: 1.09 mM salicylic acid + 100 mM glycine betaine; FC: field capacity.

**Figure 3 biomolecules-13-01634-f003:**
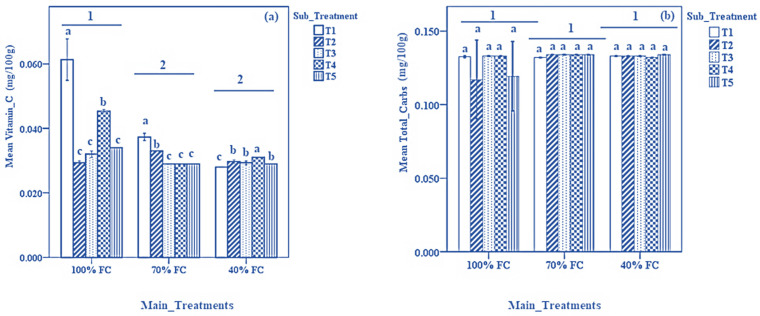
Results of (**a**): vitamin C content and (**b**): total carbohydrates content in onion bulbs under different DDI treatments and sub-treatments. Data expressed as mean ± SD. Error bars represent the standard deviation. Bar columns having the same letter under the same DDI treatment indicate no significant differences at *p* < 0.05. Horizontal lines at the top of bar columns having different numbers at the top show a significant difference between the DDI treatments at *p* < 0.05. Sub-treatments T1: control (distilled water); T2: 1.09 mM salicylic acid + 0 mM glycine betaine; T3: 1.09 mM salicylic acid + 25 mM glycine betaine; T4: 1.09 mM salicylic acid + 50 mM glycine betaine; T5: 1.09 mM salicylic acid + 100 mM glycine betaine.

**Figure 4 biomolecules-13-01634-f004:**
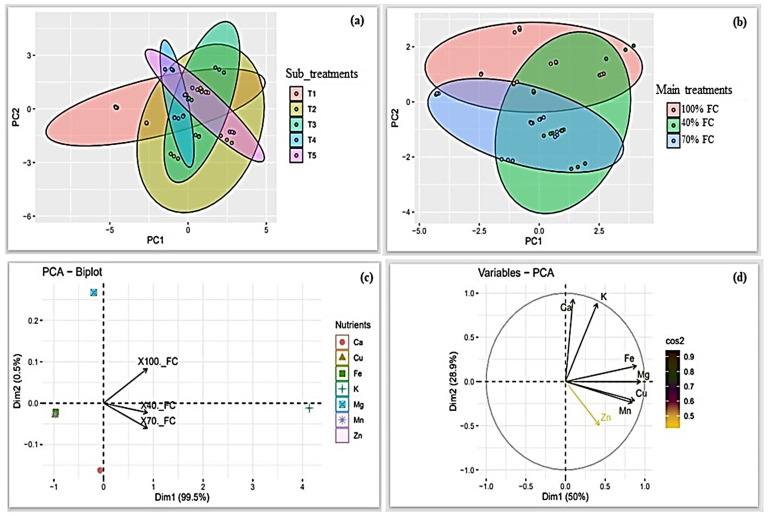
PCA biplots showing the relationships between the nutrient composition of onion bulbs and (**a**): the different sub-treatments; (**b**): the DDI treatments; (**c**): distribution of macro– and micronutrients under different DDI treatments; (**d**): variation of macro– and micronutrients. Sub–treatments T1: control (distilled water); T2: 1.09 mM salicylic acid + 0 mM glycine betaine; T3: 1.09 mM salicylic acid + 25 mM glycine betaine; T4: 1.09 mM salicylic acid + 50 mM glycine betaine; T5: 1.09 mM salicylic acid + 100 mM glycine betaine. Macro- and micronutrients K: potassium; Ca: calcium; Mg: magnesium; Mn: manganese; Cu: copper; Zn: zinc; Fe: iron.

**Table 1 biomolecules-13-01634-t001:** The physical and chemical characteristics of the experimented soil.

pH 1:2.5	EC ds/m	SP	Elements (mg/g)				Anions and Cations (meq./L)					
			N	P	K	Mn	HCO_3_^−^	Cl^−^	SO_4_^2−^	Zn^2+^	Fe^2+^	Cu^2+^
9.09	0.52	25	163	7.92	118	0.25	0.5	3.5	1.17	0.178	1.02	0.32

EC: electro conductivity, SP: particle size, N: nitrogen, P: phosphorous, K: potassium, Mn: manganese, HCO_3_**^−^**: hydrogen carbonate ions, Cl^−^: chloride ions, SO_4_^2−^: sulfate ions, Zn^2+^: zinc ions, Fe^2+^: ferrous ions, Cu^2+^: copper ions.

**Table 2 biomolecules-13-01634-t002:** Effect of exogenous application of SA alone or in combination with GB on vegetative growth of onion cultivated under different DDI treatments.

60 DAT									
Sub Treatments	Plant Height (cm)			Leaf Number/Plant			Pseudostem Diameter (mm)		
	100% FC	70% FC	40% FC	100% FC	70% FC	40% FC	100% FC	70% FC	40% FC
T1	59.73 ± 9.10 ^aB^	58.93 ± 6.95 ^aA^	53.73 ± 7.84 ^aB^	6.07 ± 1.33 ^aA^	5.93 ± 1.03 ^aA^	4.60 ± 0.74 ^aB^	19.22 ± 3.27 ^aB^	21.43 ± 3.45 ^aA^	17.45 ± 3.21 ^aB^
T2	51.97 ± 13.89 ^aB^	61.07 ± 4.89 ^aA^	54.80 ± 5.87 ^aB^	5.53 ± 1.36 ^abA^	5.60 ± 1.12 ^aA^	4.73 ± 0.80 ^aB^	17.56 ± 2.73 ^aB^	18.86 ± 2.74 ^aA^	19.25 ± 3.23 ^aB^
T3	54.83 ± 4.86 ^aB^	58.83 ± 4.82 ^aA^	52.67 ± 3.58 ^aB^	4.73 ± 0.70 ^bA^	5.93 ± 0.80 ^aA^	5.07 ± 0.70 ^aB^	15.90 ± 1.26 ^aB^	20.49 ± 3.16 ^aA^	19.88 ± 3.72 ^aB^
T4	55.73 ± 6.94 ^aB^	60.00 ± 4.69 ^aA^	50.53 ± 5.36 ^aB^	5.73 ± 1.39 ^abA^	6.00 ± 0.76 ^aA^	4.40 ± 0.63 ^aB^	17.76 ± 2.63 ^aB^	20.28 ± 2.07 ^aA^	17.00 ± 3.42 ^aB^
T5	55.93 ± 7.26 ^aB^	61.83 ± 6.70 ^aA^	52.73 ± 5.22 ^aB^	5.20 ± 0.41 ^abA^	5.73 ± 1.03 ^aA^	4.93 ± 0.88 ^aB^	16.78 ± 5.13 ^aB^	20.04 ± 3.17 ^aA^	18.07 ± 2.33 ^aB^
**75 DAT**									
T1	51.20 ± 5.54 ^aC^	58.20 ± 7.99 ^bA^	56.47 ± 6.89 ^aB^	6.73 ± 1.53 ^aB^	7.80 ± 1.82 ^aA^	6.67 ± 1.11 ^aB^	33.14 ± 6.66 ^aBC^	32.29 ± 3.71 ^aA^	32.63 ± 2.73 ^aAB^
T2	50.87 ± 4.21 ^aC^	64.07 ± 2.79 ^aA^	56.00 ± 5.45 ^aB^	6.67 ± 0.98 ^aB^	8.07 ± 1.28 ^aA^	6.80 ± 1.01 ^aB^	28.62 ± 9.78 ^aBC^	32.91 ± 2.62 ^aA^	31.40 ± 3.77 ^aAB^
T3	49.73 ± 5.70 ^aC^	59.33 ± 5.37 ^abA^	53.40 ± 13.89 ^aB^	6.07 ± 0.59 ^aB^	7.87 ± 1.06 ^aA^	6.93 ± 1.22 ^aB^	27.92 ± 3.21 ^aBC^	32.60 ± 2.93 ^aA^	28.07 ± 8.36 ^aAB^
T4	51.13 ± 5.55 ^aC^	61.07 ± 4.98 ^abA^	55.73 ± 5.26 ^aB^	6.80 ± 1.01 ^aB^	8.40 ± 1.30 ^aA^	6.47 ± 0.83 ^aB^	29.45 ± 2.62 ^aBC^	30.51 ± 3.04 ^aA^	28.19 ± 4.25 ^aAB^
T5	53.00 ± 5.94 ^aC^	63.13 ± 3.74 ^abA^	59.13 ± 3.36 ^aB^	6.47 ± 0.92 ^aB^	8.20 ± 1.32 ^aA^	6.87 ± 0.83 ^aB^	28.19 ± 3.13 ^aBC^	33.14 ± 4.41 ^aA^	30.94 ± 3.06 ^aAB^
**90 DAT**									
T1	44.47 ± 5.26 ^aC^	50.60 ± 15.58 ^bA^	46.87 ± 10.01 ^aB^	7.67 ± 1.68 ^aB^	8.00 ± 2.85 ^aA^	8.27 ± 1.98 ^aB^	32.63 ± 4.64 ^aB^	37.09 ± 4.58 ^aA^	32.58 ± 4.63 ^aB^
T2	44.13 ± 4.73 ^aC^	59.20 ± 4.68 ^aA^	46.73 ± 3.32 ^aB^	7.80 ± 2.08 ^aB^	9.27 ± 1.58 ^aA^	7.80 ± 1.74 ^aB^	31.16 ± 6.94 ^aB^	35.50 ± 4.59 ^aA^	32.27 ± 3.95 ^aB^
T3	42.53 ± 3.66 ^aC^	56.60 ± 3.68 ^abA^	45.80 ± 6.14 ^aB^	7.33 ± 1.40 ^aB^	8.73 ± 1.62 ^aA^	7.27 ± 1.79 ^aB^	29.99 ± 5.00 ^aB^	38.56 ± 5.26 ^aA^	31.74 ± 3.96 ^aB^
T4	44.67 ± 5.04 ^aC^	56.93 ± 5.08 ^abA^	46.87 ± 5.64 ^aB^	8.33 ± 1.05 ^aB^	8.93 ± 1.28 ^aA^	7.06 ± 1.39 ^aB^	31.71 ± 5.29 ^aB^	34.97 ± 2.93 ^aA^	29.17 ± 4.45 ^aB^
T5	43.53 ± 4.70 ^aC^	58.67 ± 4.12 ^abA^	48.73 ± 4.03 ^aB^	7.00 ± 1.13 ^aB^	9.80 ± 1.52 ^aA^	7.67 ± 1.23 ^aB^	30.59 ± 8.06 ^aB^	36.23 ± 3.63 ^aA^	32.23 ± 5.22 ^aB^

Data expressed as mean ± SD. Different lower superscript letters within the same column indicate a significant difference within the sub-treatments at *p* < 0.05. Different upper superscript letters within the different columns indicate a significant difference between the DDI treatments at *p* < 0.05. Sub-treatments T1: control (distilled water); T2: 1.09 mM salicylic acid + 0 mM glycine betaine; T3: 1.09 mM salicylic acid + 25 mM glycine betaine; T4: 1.09 mM salicylic acid + 50 mM glycine betaine; T5: 1.09 mM salicylic acid + 100 mM glycine betaine; DAT: days after transplanting; FC: field capacity.

**Table 3 biomolecules-13-01634-t003:** Effect of exogenous application of SA alone or in combination with GB on bulb quality of onion cultivated under different DDI treatments.

Sub-Treatments	% Doubles			% Grade A (<50 mm)			% Grade B (50–70 mm)			% Grade C (>70 mm)		
	100% FC	70% FC	40% FC	100% FC	70% FC	40% FC	100% FC	70% FC	40% FC	100% FC	70% FC	40% FC
**T1**	13.81 ± 11.98 ^aB^	42.42 ± 5.25 ^aA^	26.11 ± 21.50 ^aAB^	2.78 ± 4.80 ^aA^	0.00 ± 0.00 ^aAB^	0.00 ± 0.00 ^aB^	47.22 ± 4.81 ^aA^	18.18 ± 9.09 ^aB^	36.11 ± 12.73 ^aA^	50.00 ± 0.00 ^aC^	81.81 ± 9.09 ^aA^	58.33 ± 8.34 ^aB^
**T2**	18.84 ± 11.00 ^aB^	34.20 ± 12.08 ^aA^	23.33 ± 8.82 ^aAB^	5.59 ± 4.89 ^aA^	7.14 ± 12.37 ^aAB^	2.78 ± 4.81 ^aB^	42.54 ± 17.00 ^aA^	16.23 ± 10.20 ^aB^	35.00 ± 6.01 ^aA^	51.86 ± 16.51 ^aC^	79.00 ± 6.51 ^aA^	62.20 ± 6.74 ^aB^
**T3**	13.59 ± 15.25 ^aB^	35.84 ± 10.85 ^aA^	34.40 ± 26.62 ^aAB^	2.78 ± 4.80 ^aA^	3.03 ± 5.25 ^aAB^	0.00 ± 0.00 ^aB^	59.61 ± 13.46 ^aA^	10.93 ± 3.87 ^aB^	29.91 ± 5.93 ^aA^	37.61 ± 18.23 ^aC^	86.03 ± 5.08 ^aA^	75.64 ± 8.40 ^aB^
**T4**	25.12 ± 2.10 ^aB^	39.74 ± 9.68 ^aA^	21.15 ± 19.52 ^aAB^	11.62 ± 10.09 ^aA^	0.00 ± 0.00 ^aAB^	2.56 ± 4.44 ^aB^	27.68 ± 2.91 ^aA^	15.02 ± 0.63 ^aB^	40.23 ± 21.45 ^aA^	60.70 ± 7.62 ^aC^	84.98 ± 0.63 ^aA^	57.20 ± 18.66 ^aB^
**T5**	23.89 ± 20.16 ^aB^	42.86 ± 26.53 ^aA^	33.89 ± 25.84 ^aAB^	7.22 ± 6.73 ^aA^	0.00 ± 0.00 ^aAB^	0.00 ± 0.00 ^aB^	42.78 ± 23.35 ^aA^	30.43 ± 18.67 ^aB^	31.11 ± 18.28 ^aA^	50.00 ± 22.05 ^aC^	69.56 ± 18.67 ^aA^	71.67 ± 16.42 ^aB^

Data expressed as mean ± SD. Different lower superscript letters within the same column indicate a significant difference within the sub-treatments at *p* < 0.05. Different upper superscript letters within different columns indicate a significant difference between the DDI treatments at *p* < 0.05. Sub-treatments T1: control (distilled water); T2: 1.09 mM salicylic acid + 0 mM glycine betaine; T3: 1.09 mM salicylic acid + 25 mM glycine betaine; T4: 1.09 mM salicylic acid + 50 mM glycine betaine; T5: 1.09 mM salicylic acid + 100 mM glycine betaine; FC: field capacity.

**Table 4 biomolecules-13-01634-t004:** Effect of exogenous application of SA alone or in combination with GB on macronutrient composition of onion cultivated under different DDI treatments.

Sub-Treatments	K (mg/100 g)			Ca (mg/100 g)			Mg (mg/100 g)		
	100% FC	70% FC	40% FC	100% FC	70% FC	40% FC	100% FC	70% FC	40% FC
**T1**	836.87 ± 2.15 ^dC^	2293.97 ± 1.40 ^eB^	8108.17 ± 0.35 ^cA^	336.43 ± 1.70 ^bC^	419.27 ± 0.64 ^eA^	1206.67 ± 4.04 ^cB^	506.72 ± 0.35 ^eB^	243.79 ± 0.38 ^eC^	690.59 ± 0.75 ^eA^
**T2**	396.83 ± 2.10 ^eC^	7699.60 ± 0.36 ^bB^	2824.20 ± 0.00 ^eA^	301.07 ± 0.29 ^cC^	1599.23 ± 0.29 ^bA^	364.80 ± 3.80 ^eB^	524.84 ± 0.13 ^dB^	860.37 ± 0.58 ^aC^	1130.06 ± 0.52 ^aA^
**T3**	2539.07 ± 1.53 ^cC^	5773.97 ± 0.97 ^dB^	10,743.53 ± 3.55 ^aA^	224.97 ± 2.91 ^eC^	1425.23 ± 2.27 ^cA^	2226.10 ± 3.84 ^aB^	829.26 ± 0.45 ^bB^	777.09 ± 2.59 ^bC^	799.02 ± 0.05 ^cA^
**T4**	2981.57 ± 1.65 ^bC^	8155.93 ± 5.46 ^aB^	3734.23 ± 0.23 ^dA^	829.30 ± 0.95 ^aC^	2102.93 ± 2.32 ^aA^	734.47 ± 0.81 ^dB^	548.22 ± 0.30 ^cB^	587.62 ± 0.59 ^dC^	767.65 ± 0.99 ^dA^
**T5**	6722.47 ± 0.60 ^aC^	7461.73 ± 0.64 ^cB^	8315.13 ± 2.86 ^bA^	249.43 ± 0.40 ^dC^	1070.43 ± 0.65 ^dA^	1491.99 ± 0.46 ^bB^	950.98 ± 0.75 ^aB^	650.89 ± 0.39 ^cC^	841.51 ± 0.46 ^bA^

Data expressed as mean ± SD. Different lower superscript letters within the same column indicate a significant difference within the sub-treatments at *p* < 0.05. Different upper superscript letters within different columns indicate a significant difference between the DDI treatments at *p* < 0.05. Sub-treatments T1: control (distilled water); T2: 1.09 mM salicylic acid + 0 mM glycine betaine; T3: 1.09 mM salicylic acid + 25 mM glycine betaine; T4: 1.09 mM salicylic acid + 50 mM glycine betaine; T5: 1.09 mM salicylic acid + 100 mM glycine betaine; FC: field capacity.

**Table 5 biomolecules-13-01634-t005:** Effect of exogenous application of SA alone or in combination with GB on micronutrient composition of onion cultivated under different DDI treatments.

Sub-Treatments	Mn (mg/100 g)			Cu (mg/100 g)			Zn (mg/100 g)			Fe (mg/100 g)		
	100% FC	70% FC	40% FC	100% FC	70% FC	40% FC	100% FC	70% FC	40% FC	100% FC	70% FC	40% FC
**T1**	3.46 ± 0.02 ^cB^	1.66 ± 0.02 ^cC^	6.51 ± 0.47 ^bA^	1.38 ± 0.02 ^eA^	0.53 ± 0.02 ^aB^	2.19 ± 0.18 ^aA^	7.75 ± 0.25 ^dA^	3.53 ± 0.66 ^dC^	12.29 ± 0.29 ^dB^	12.45 ± 0.15 ^dB^	6.29 ± 0.51 ^dB^	30.41 ± 0.85 ^aA^
**T2**	5.14 ± 0.11 ^bB^	5.46 ± 0.06 ^aC^	13.01 ± 1.41 ^aA^	1.64 ± 0.08 ^dA^	1.74 ± 0.05 ^bB^	3.21 ± 0.70 ^aA^	43.15 ± 2.94 ^aA^	19.28 ± 2.63 ^aC^	22.02 ± 0.48 ^aB^	15.36 ± 1.17 ^cB^	26.29 ± 0.51 ^aB^	30.15 ± 1.05 ^aA^
**T3**	6.03 ± 0.05 ^bB^	4.66 ± 0.58 ^bC^	7.07 ± 0.09 ^bA^	2.72 ± 0.06 ^bA^	2.10 ± 0.13 ^aB^	2.54 ± 0.45 ^aA^	18.02 ± 1.01 ^cA^	17.35 ± 0.82 ^abC^	15.31 ± 0.63 ^bB^	23.39 ± 1.01 ^bB^	21.43 ± 0.53 ^bB^	28.69 ± 0.57 ^aA^
**T4**	7.38 ± 0.63 ^aB^	4.03 ± 0.56 ^bC^	4.60 ± 0.67 ^cA^	2.32 ± 0.01 ^cA^	1.17 ± 0.14 ^cB^	2.09 ± 0.04 ^aA^	10.41 ± 0.51 ^dA^	11.24 ± 0.29 ^cC^	14.65 ± 0.48 ^bcB^	9.53 ± 0.40 ^eB^	15.62 ± 2.56 ^cB^	21.25 ± 0.40 ^bA^
**T5**	7.59 ± 0.47 ^aB^	5.80 ± 0.18 ^aC^	5.47 ± 0.51 ^bcA^	3.63 ± 0.05 ^aA^	1.33 ± 0.09 ^cB^	2.58 ± 0.47 ^aA^	25.04 ± 0.99 ^bA^	15.49 ± 0.46 ^bC^	13.43 ± 0.62 ^cdB^	30.05 ± 0.10 ^aB^	22.87 ± 0.86 ^bB^	18.37 ± 0.72 ^cA^

Data expressed as mean ± SD. Different lower superscript letters within the same column indicate a significant difference within the sub-treatments at *p* < 0.05. Different upper superscript letters within different columns indicate a significant difference between the DDI treatments at *p* < 0.05. Sub-treatments T1: control (distilled water); T2: 1.09 mM salicylic acid + 0 mM glycine betaine; T3: 1.09 mM salicylic acid + 25 mM glycine betaine; T4: 1.09 mM salicylic acid + 50 mM glycine betaine; T5: 1.09 mM salicylic acid + 100 mM glycine betaine; FC: field capacity.

## Data Availability

The data presented in this study are available in this article.

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
