# Peer review of "Coping with Water Stress: Ameliorative Effects of Combined Treatments of Salicylic Acid and Glycine Betaine on the Biometric Traits and Water-Use Efficiency of Onion (Allium cepa) Cultivated under Deficit Drip Irrigation"

_biomolecules, 2023, doi:10.3390/biom13111634_

Round 1
Reviewer 1 Report
Comments and Suggestions for Authors
Extremely interesting for practice. It needs to publish. However the accent for the agriculture problem does not exclude some words about mechanisms of used SA and GB on plant cell in order to understand of possible nature of their effects. List of abbreviations before the Introduction section will be more suitable for readers.
Reviewer 2 Report
Comments and Suggestions for Authors
The manuscript entitled “Coping with water stress: Ameliorative effects of combined treatments of salicylic acid and glycine betaine on the biometric traits and water use efficiency of onion (Allium cepa) cultivated under deficit drip irrigation” signed by Muziri Mugwanya et al. presents interesting results regarding the investigation of the influence of application of salicylic acid alone or in combination with glycine betaine on the growth, yield quality and other properties on different deficit drip irrigation treatments. The topic is highly actual and the results could be of high interest in the literature, however, I have some of aspects that need clarification and are mentioned below:
Major issue: Were the onions dried (freeze-dried) before analysis? I guess not. Otherwise, the water content is required and could be determined so that any reader could convert the values expressed in fresh material to dried material. Also, water content variation might be explains some variations in the chemical analysis.
Abstract: explain what FC acronym means directly in the abstract. Also, the abstract must contain not only information about the methodology which it does but also specific information regarding the results, findings, and explanations. There are only general information regarding the results, but the authors need to introduce the main specific results directly in the abstract.
Introduction: Generally, well written and clear. It covers the all the required information in this context.
Materials and methods: It contains eight subsections. Generally, they are well written, I a correct way and they contain enough details in order to be reproduced.
-Since the treatment did contain tween 20 as wetting agent, why the authors did not consider the control not simple as water but this tween 20 at the same concentration as in the treatments?
Results: The section contains five subsections, very well organized
Discussions: Clearly written
